# Optimizing Invasive Neonatal Respiratory Care: A Systematic Review of Invasive Neurally Adjusted Ventilatory Assist

**DOI:** 10.3390/healthcare12060632

**Published:** 2024-03-11

**Authors:** Palanikumar Balasundaram, Mohamed Sakr

**Affiliations:** 1Division of Neonatology, Department of Pediatrics, Mercy Health—Javon Bea Hospital, Rockford, IL 61114, USA; 2Division of Neonatology, The Children’s Hospital at Montefiore, Albert Einstein College of Medicine, Bronx, NY 10461, USA; msakr@montefiore.org

**Keywords:** neurally adjusted ventilatory assist, bronchopulmonary dysplasia, preterm, patient–ventilator synchrony

## Abstract

Background: Mechanical ventilation in preterm neonates aims for synchrony, preventing complications such as lung injury. Neurally Adjusted Ventilatory Assist (NAVA) is a unique mode relying on diaphragmatic electrical signals for synchronization. We conducted a review focusing on the long-term consequences of using invasive NAVA in neonates with a focus on bronchopulmonary dysplasia (BPD). Methods: A systematic review following PRISMA explored invasive NAVA in preterm neonates. Primary objectives compared NAVA to conventional ventilation, assessing BPD incidence, ventilation duration, length of stay, and adverse events. Secondary objectives analyzed ventilator parameters. Results: After screening 282 records, the review incorporated two randomized controlled trials for primary outcomes and seven trials for secondary outcomes, including two randomized crossovers, four prospective crossovers, and one retrospective study. NAVA showed reduced oxygen requirement at 28 days but no significant differences in oxygen need at 36 weeks postmenstrual age, total length of stay, or ventilator days. Substantial variations were not observed in adverse events. Ventilator variables favored NAVA, indicating decreased peak inspiratory pressure, tidal volume, work of breathing, and respiratory severity score. Conclusion: Our study found no significant reduction in BPD with NAVA despite short-term benefits. Future large-scale trials are essential to assess NAVA’s impact on long-term outcomes comprehensively.

## 1. Introduction

Ensuring patient–ventilator synchrony is one of the main objectives of mechanical ventilation in preterm neonates, particularly those with Extremely Low Birth Weight (ELBW < 1000 g). In critically ill pediatric, adult, and animal studies, patient–ventilator asynchrony has been linked to increased work of breathing, lung injury, diaphragmatic dysfunction, and mortality [1,2,3,4,5,6,7,8,9,10,11,12,13]. Asynchrony also has negative consequences linked to patient discomfort and poor sleep quality [14,15,16]. Many factors contribute to asynchrony in preterm neonates, including high variability in breathing patterns, high respiratory rate, small tidal volume, and leaks, which are inevitable in invasive mechanical ventilation in neonates owing to the use of uncuffed endotracheal tubes [17,18]. Several types of asynchronies have been identified, which are those related to triggering (auto, ineffective, double, or reverse triggering), those related to cycling (premature or delayed cycling), and those related to flow (excessive or insufficient flow) [19]. Although studies have shown that noninvasive respiratory support is superior [20,21], invasive mechanical ventilation is unavoidable, particularly in preterm neonates born at less than 28 weeks of gestation [22,23]. Various ventilatory modes are currently used in neonatal mechanical ventilation, with Neurally Adjusted Ventilatory Assist (NAVA) standing out due to its unique and exceptionally appealing characteristics.

NAVA is a form of respiratory support that relies on the electromyographic potential of the diaphragm to trigger the ventilator rather than pneumatic pressure or a flow signal located at the airway opening or the ventilator end [24,25,26]. This unique property of NAVA minimizes patient–ventilatory asynchrony through several mechanisms. NAVA ensures synchronization not only during breath initiation but also during breath termination. The magnitude of the patient’s respiratory demand on NAVA is synchronized between the patient and the ventilator [24,25,26]. Several clinical studies in neonates have demonstrated a significant improvement in synchrony between the patient and the ventilator when utilizing NAVA, whether invasive or noninvasive [17,18,27,28,29,30]. Moreover, up to the present moment, the Electrical Activity of the Diaphragm (EAdi) represents the earliest and foremost signal in the neural respiratory pathway (beginning from the central respiratory system, traversing the phrenic nerve, and concluding with diaphragmatic contraction), in contrast to changes in pressure or flow, which is, in fact, the last event in the neuro-respiratory course [25]. Additionally, reliance on flow or pressure sensors can be affected by leakage in the ventilator circuit and thus reduces the accuracy of both the monitoring and delivery of ventilatory assistance [17,18,27,28,29,30].

NAVA delivers pressure assistance proportional to and synchronized with the EAdi. A specially designed single-use feeding tube with measuring electrodes is positioned in the esophagus, enabling the isolation of diaphragmatic electrical signals from other signals in the body, especially those originating from the heart [25,26,31,32]. The NAVA level is the factor by which the Edi signal is multiplied to adjust the amount of ventilator assist delivered. Utilizing the Edi waveform, NAVA can monitor neural respiratory drive and breathing patterns, providing valuable insights even without the routine techniques such as measuring airflow, airway pressure, or volume commonly employed in conventional mechanical ventilation (CMV) [33,34]. NAVA can provide ventilator assistance that is synchronized in relation to triggering, also cycling the magnitude of the patient’s breath. In addition, observing the EAdi and pressure waveforms can provide a real-time assessment of the degree of synchronization [35].

Bronchopulmonary dysplasia (BPD), commonly referred to as chronic lung disease, is a significant complication observed in preterm neonates. BPD is a multifactorial disease of preterm neonates and stands as one of the leading causes of morbidity and mortality in these populations. The etiology of BPD is influenced by various antenatal and postnatal factors, among which ventilator-induced lung injury emerges as a significant contributor to the pathogenesis of this condition. The prevalence of BPD continues to be elevated and has shown minimal alteration over the past decade. The most effective strategy for reducing the incidence of BPD is preventing premature birth; however, if this is not achievable, it becomes crucial to minimize lung injury. While there is a widespread agreement that steering clear of invasive ventilation is an evidence-based approach to mitigate the risk of BPD, over one-third of preterm neonates receive invasive respiratory support during their hospitalization [36]. Synchronizing neonates’ spontaneous inspiratory efforts with positive-pressure inflations enhances ventilation effectiveness at lower inflation pressures, minimizing the risk of lung injury. Notably, NAVA has demonstrated feasibility in practical use among ELBW neonates [17].

Several studies have compared NAVA to other conventional mechanical ventilator modes and have mainly been centered on the short-term outcomes, including respiratory support parameters, lung mechanics, Work of Breathing (WOB), duration of mechanical ventilation, sedation requirements, and EAdi [37,38,39,40,41,42,43,44,45,46,47,48]. However, only a few studies have investigated the long-term outcomes of applying this novel technique in the preterm population. We attempted to review studies that have explored the long-term consequences of using NAVA in neonates with a specific focus on respiratory outcomes, namely BPD. The objective of our research question delves into the nuanced impact of NAVA on the incidence of BPD and assesses the risks such as mortality risk, intraventricular hemorrhage, air leak, and patent ductus arteriosus, as well as the duration of mechanical ventilation and total length of stay. The hypothesis driving this systematic review posits that NAVA may offer advantages over conventional ventilation methods, potentially leading to reduced incidences of BPD and associated complications in preterm newborns. By synthesizing evidence from diverse studies, we aim to provide a comprehensive understanding of the long-term effects of NAVA, contributing valuable insights to guide future clinical practices and interventions in neonatal care. 

## 2. Materials and Methods

### 2.1. Criteria for Considering Studies for This Review

A systematic review was conducted following the guidelines outlined by the Preferred Reporting Items for Systematic Reviews and Meta-Analyses (PRISMA) [49] to explore the application of invasive NAVA in preterm newborn infants (Appendix A). The primary objectives included investigating whether NAVA leads to lower incidences of BPD compared to conventional triggered ventilation methods, as well as assessing the risks for mortality, intraventricular hemorrhage (IVH), air leak, patent ductus arteriosus (PDA), duration of mechanical ventilation, and total length of stay in NAVA and conventional ventilation groups. In addition to primary objectives, secondary objectives involved examining differences in various ventilator parameters, including peak inspiratory pressure (PIP), mean airway pressure (MAP), Tidal Volume (TV), Work of Breathing (WOB), and Respiratory Severity Score (RSS) between NAVA and conventional ventilation from crossover trials in newborns.

### 2.2. Search Methods for the Identification of Studies

A comprehensive literature search strategy encompassed various databases, including PubMed, Embase, Google Scholar, clinicaltrials.gov, and the Cochrane Library. The search strategy incorporated Medical Subject Headings (MeSH) terms like ‘infant’, ‘newborn’, and ‘Interactive Ventilatory Support’ alongside title keywords such as ‘neurally adjusted’ and ‘NAVA’ without constraints on date, language, or publication status. The strategic application of Boolean operators ‘AND’ and ‘OR’ enhanced the search to refine and broaden the relevant literature in the field. Two independent assessors were involved in the literature review process. In the event of a disagreement, efforts were made to discuss the discrepancy and reach a consensus through further discussion of the relevant literature. The systematic review has been registered prospectively under PROSPERO with ID CRD42024506598 to ensure transparency and adherence to established protocols. 

### 2.3. Data Collection and Data Analysis

#### 2.3.1. Selection of Studies

The inclusion criteria for this review encompass randomized controlled trials (RCTs) or crossover designs involving preterm infants necessitating invasive ventilator support, focusing on comparing NAVA with CMV. The study aims to explore oxygen needs at 36 weeks postmenstrual age as the primary outcome, gauging NAVA’s effectiveness in reducing BPD instances. Secondary outcomes include evaluating oxygen needs at 28 days, ventilator days, length of stay, and adverse events. The review will also explore ventilator variables from newborn crossover trials. Exclusion criteria were applied to ensure the relevance and reliability of the selected studies. Non-English articles and those involving populations beyond preterm infants were excluded. Studies focusing on interventions other than invasive NAVA were excluded, specifically excluding non-invasive NAVA studies. Additionally, studies lacking control arms were excluded to enhance the rigor of the analysis. Conference abstracts, case reports, case reviews, and narrative reviews were also excluded to maintain a focus on high-quality empirical research for this review.

#### 2.3.2. Data Extraction and Management

Two reviewers independently extracted data from the studies, encompassing study characteristics and outcomes, including publication year and author information. 

#### 2.3.3. Data Synthesis

For outcomes, the difference between the means of the data after the intervention period was extracted. In cases where the average was unavailable, the difference between the average of the changed values and their standard deviation was extracted. Effect measures were chosen based on the nature of the variables. Relative risk ratios (RRs) with 95% confidence intervals (CI) served as the effect measure for binary variables such as O_2_ requirement at 28 days, O_2_ requirement at 36 weeks, and adverse events (mortality rate, PDA, IVH, air leak). Mean difference (MD), standardized MD (SMD), and 95% CI were used as the effect measures for continuous variables, including the duration of invasive ventilation, length of stay in the Neonatal Intensive Care Unit (NICU), and ventilator variables (PIP, MAP, TV, RSS, WOB). The Respiratory Severity Score (RSS) is determined by the product of MAP and the Fraction of Inspired Oxygen (FiO_2_). RSS serves as a quantitative measure of the severity of respiratory conditions. The RSS has been employed as an alternative to the oxygenation index for assessing the need for respiratory support in infants on assisted ventilation, and some studies have demonstrated robust correlations between RSS and subsequent severe BPD and mortality in preterm neonates [50]. The Work of Breathing (WOB) measures the energy required to achieve inspiratory pressure above positive end-expiratory pressure. WOB is determined by integrating the product of pressure and volume changes for each breath, visualized as the area under the pressure–volume curve [51]. The data acquisition program of the ventilator in operation is utilized to determine and quantify the WOB, providing valuable insights into the respiratory effort exerted by the patient. Using NCSS 2023 Statistical Software (NCSS, LLC, Kaysville, UT, USA), we generated comprehensive summary statistics and forest plots for these variables, contributing to a robust understanding of outcomes associated with these distinctions. 

#### 2.3.4. Quality of Evidence

To gauge the quality of the chosen articles and assess potential biases, we employed the Scottish Intercollegiate Guidelines Network (SIGN) framework, which categorizes evidence levels and assigns recommendation grades [52]. The levels of evidence range from 1 to 4, with corresponding symbols such as “++”, “+”, and “−” for scores 1 and 2, providing additional granularity to the evidence level. Additionally, the SIGN guide facilitates the assignment of grades A to D, with a direct recommendation resulting in a classification of “A”. Two team members independently conducted the final selection of studies for inclusion following the initial classification to mitigate individual bias. 

#### 2.3.5. Assessment of Heterogeneity

The heterogeneity within the selected studies was examined using the I2 statistic and Cochran’s Q statistic. I2 statistics provide insights into the degree of heterogeneity, with interpretations ranging from absent (0) to low (25), moderate (50), or high (75 or higher). A *p*-value below 0.1 for Cochran’s Q statistic indicates statistically significant heterogeneity [53].

#### 2.3.6. Assessment of Risk of Bias

The Cochrane Risk of Bias 2.0 (RoB2) tool was employed to assess bias in RCTs and studies with random assignment. The RoB2 tool evaluated five domains, namely randomization, deviations from interventions, missing data, outcome measurement, and reported outcomes. Bias within each outcome was categorized as low risk, some concerns, or high risk [54]. The ROBINS-I tool, version 1—2016 (Risk of Bias in Non-Randomized Studies of Interventions) was employed to assess bias in non-randomized intervention studies across seven domains, namely confusion, participant selection, intervention classification, deviations from stipulated interventions, data completeness, variable measurement, and outcome selection. Each domain was rated low, moderate, serious, critical, or no information risk [55].

## 3. Results

### 3.1. Study Selection

In December 2023, we conducted a thorough search for pertinent articles. A total of 282 records were initially identified through database searching. After excluding duplicates (*n =* 196), 86 records underwent screening. Subsequently, 26 reports were considered for full-text articles, but two were not retrieved. Following the eligibility assessment of 24 reports, 15 were excluded, resulting in the final inclusion of 9 studies in the review. Figure 1 provides an overview of our search methods and the study selection process. 

### 3.2. Study Characteristics

Of the publications finally selected for analysis, only two studies (Fang et al. and Kallio et al.) addressed the primary objective, focusing on the incidence of BPD and adverse events associated with ventilators [56,57]. Fang et al. and Kallio et al. assessed oxygen requirement at 36 weeks postmenstrual age (PMA), but only Fang et al. assessed oxygen requirement at 28 days of life. Additionally, both Fang et al. and Kallio et al. evaluated ventilator days and total length of stay. Among the studies included in the review, seven studies [41,48,58,59,60,61,62] addressed ventilator variables, our secondary objective. Out of those seven trials, two were randomized crossovers, four were prospective crossovers, and one was a retrospective study. The characteristics of these included studies are detailed in Table 1. 

### 3.3. Level of Evidence and Grade of Recommendation

Table 2 summarizes the levels of evidence and grades of recommendation for the studies included in the review as per the SIGN framework. 

### 3.4. Methodological Quality and Risk of Bias

The RoB2 tool [54] revealed an overall outcome indicating “some concerns” regarding bias in two RCT studies and a “high risk” in two randomized crossover studies (Appendix A). Two of the five non-randomized studies assessed using the ROBINS-I tool [55] exhibited an overall serious risk, whereas the remaining three showed a moderate risk of bias (Appendix A).

### 3.5. Impact on BPD Outcome 

The National Institute of Child Health and Human Development (NICHD) revised BPD criteria, defining it as the need for oxygen support (>21%) at 36 weeks postmenstrual age (PMA) for infants <32 weeks and for >28 days for those >32 weeks [63]. The Vermont Oxford Network (VON) defines BPD based solely on FiO_2_ requirement at 36 weeks PMA [64]. Two studies, Fang et al. and Kallio et al., reported data on O_2_ requirements at 36 weeks PMA, with Fang et al. also including information on O_2_ requirements at 28 days of life.

The combined analysis, encompassing studies by Fang et al. and Kallio et al., comparing proportions regarding oxygen requirements at 36 weeks PMA for NAVA vs. CMV, reveals a combined risk ratio of 0.8143 (95% CI: 0.4055 to 1.6351). However, the non-directional zero-effect test (*p* = 0.3729) and directional zero-effect test (*p* = 0.5636) did not achieve statistical significance. Additionally, the heterogeneity test (Cochran’s Q = 1.6396, *p* = 0.2004) suggested homogeneous effects across studies, further supporting the consistency of the findings (Figure 2).

The analysis comparing oxygen requirement at 28 days for NAVA versus CMV, based on the study by Fang et al., revealed a risk ratio of 0.6029 (95% CI: 0.3827 to 0.9498). The random effects model suggests a significant directional effect (*p* = 0.0291), indicating that patients on NAVA have a reduced risk of needing oxygen at 28 days compared to CMV. 

### 3.6. Impact on Length of Stay and Ventilator Days

When comparing the total length of NICU stay between NAVA and conventional models, the combined analysis showed a fixed mean difference of −1.395 (95% CI: −3.308 to 0.517), suggesting a potential decrease in total NICU stay length with NAVA compared to conventional models, as implied by the negative fixed mean difference of −1.395. However, it is crucial to approach this finding with caution due to the lack of statistical significance and notable heterogeneity among the included studies (τ² = 48.036) (Figure 3).

Similarly, in comparing ventilator days between NAVA and conventional models, the combined analysis suggested a potential decrease with NAVA, but the observed average difference of −4.8228 (95% CI: −14.5223 to 4.8767) was not statistically significant. Caution is warranted due to the significant heterogeneity observed among the included studies (τ² = 48.741) (Figure 4). Further research and consideration of individual study characteristics are necessary to draw more definitive conclusions on the impact of NAVA on NICU stay length and ventilator days.

### 3.7. Impact on Adverse Events

The combined analysis comparing NAVA and conventional ventilation suggests a non-significant decrease in the risk of air leaks with NAVA (risk ratio: 0.7301, 95% CI: 0.2186 to 2.4387, *p* = 0.6092). There is a non-significant increase in the risk of IVH with NAVA (risk ratio: 2.0024, 95% CI: 0.7390 to 5.4254, *p* = 0.6135). No significant difference is observed in death rates (risk ratio: 1.5681, 95% CI: 0.2736 to 8.9871, *p* = 0.1884) and PDA incidence (risk ratio: 0.6198, 95% CI: 0.3039 to 1.2642, *p* = 0.2551). Heterogeneity tests indicate no significant variation in effects among studies for all comparisons.

### 3.8. Impact on Ventilator Variables

The combined analysis indicates a significant decrease in peak inspiratory pressure (PIP) in NAVA versus CMV, with a mean difference of −3.367 (95% CI: −4.327 to −2.407), across studies including Hunt et al., Jung et al., Lee et al., Rosterman et al., and Stein et al. For tidal volume (TV) in NAVA versus CMV, a significant mean difference of −0.738 (95% CI: −1.139 to −0.338) is observed, with both fixed and random-effect models showing significance and an inconsistency index (I^2^) of 72.065%. In the comparison of mean airway pressures (MAP), a non-significant mean difference of −0.264 (95% CI: −0.631 to 0.104) is found between NAVA and CMV, but there is significant overall heterogeneity (I^2^ = 83.677%). NAVA demonstrates a significantly lower Work of Breathing (WOB) compared to CMV, with a mean difference of −0.331 (95% CI: [−0.410, −0.251]). Despite statistical significance, notable heterogeneity exists among the studies (Cochran’s Q test, Q = 9.300, *p* = 0.0023). The combined analysis comparing NAVA and CMV for Respiratory Severity Score (RSS) shows a significant reduction with NAVA (mean difference −1.179, 95% CI: [−1.772, −0.587]). However, heterogeneity exists among studies (Cochran’s Q = 5.993, *p* = 0.0144) (Appendix A).

## 4. Discussion

Based on the limited number of studies exploring the long-term effects of invasive NAVA in neonates, this meta-analysis is the first to address this knowledge gap in the literature. The prevalence of BPD continues to be substantial despite significant advancements in neonatal care and contemporary ventilatory management [20,65,66,67,68]. Compared to CMV, NAVA has been found to improve patient–ventilator interaction and minimize asynchrony [2,17,18,28,35]. Improving patient–ventilator synchrony holds promise for the care of preterm neonates, as it has the potential to mitigate lung injury and may play a critical role in reducing the burden of BPD.

In this systematic review and meta-analysis, the comparative assessment of invasive NAVA and CMV did not reveal a superiority of NAVA concerning clinically relevant outcomes for premature infants. Despite observing a decreased risk of oxygen requirement at 28 days with NAVA compared to CMV, the combined analysis of RCTs did not reveal a statistically significant reduction in oxygen requirements at 36 weeks PMA. Additionally, the combined meta-analysis could not establish a statistically significant difference between the two ventilator modalities concerning length of stay, ventilator days, IVH, air leak, PDA, or mortality. With the limited number of available RCTs and the observed bias among the analyzed studies, generating a definitive recommendation regarding the use of invasive NAVA versus other CMV modes may be constrained by the current body of evidence. Rossor et al. [69] incorporated only one study in their systematic review, discovering no significant differences in the duration of mechanical ventilation or the rates of BPD. Similar outcomes were observed in our broader systematic review encompassing more studies.

The analysis of ventilator variables demonstrated a significant decrease in PIP, TV, WOB, and RSS with NAVA compared to conventional ventilation, while no significant difference was observed in MAP. Additionally, NAVA demonstrated lower WOB and RSS. These findings underscore favorable ventilator variables associated with the NAVA mode. However, it is important to interpret these findings cautiously, as significant overall heterogeneity suggests variability among the studies.

In our comparative analysis, Fang et al. [56] and Kallio et al. [57] emerged as crucial contributors, each shedding a unique light on the use of invasive NAVA in neonatal ventilation. Fang et al. [56] focused on infants with a GA < 32 weeks who were intubated during delivery room resuscitation and were enrolled within 24 h of birth. On the other hand, Kallio et al. [57] encompassed a broader GA range (28 to 36 6/7 weeks) and concentrated on infants with RDS who were intubated for at least 4 h, with a median enrollment age of 9 days. These distinctions in enrollment criteria highlight the diverse neonatal populations studied in this review.

Although both trials used patient-triggered pressure-limited ventilation in the control group, they used two different ventilator modes. Kallio et al. [57] used pressure-controlled ventilation (PCV), whereas Fang et al. [56] used either synchronized intermittent mandatory ventilation (SIMV) or SIMV with pressure support (SIMV-PS). These modes support the infant’s breathing differently, but none have been shown to be superior except for in the duration of weaning, which tended to be shorter in infants supported by PCV [70]. Hence, using different modes of pressure-limited ventilation in these studies is not the reason for the difference in outcomes. 

Multiple studies investigated sedation medication usage and pain scores during NAVA ventilation. All these studies have shown that sedation requirements were lower in the NAVA group in comparison to CMV. Notably, these studies used different pain scales such as the PIPP (Premature Infant Pain Profile) scale or the FLACC (Face, Legs, Activity, Cry, and Consolability) scale and different pain and sedative medications. Moreover, the patient’s characteristics were very different regarding gestational age, postnatal age, birth weight, respiratory pathology, and severity of the respiratory status at the time of NAVA application. Thus, it might be difficult, with the current evidence, to reach an ultimate conclusion regarding the use of sedatives and pain management during NAVA ventilation [37,45,46].

The crossover studies included in the review for analyzing ventilator variables vary in their designs, sample sizes, patient characteristics, and inclusion–exclusion criteria, contributing to the diversity in the reported outcomes. Regarding study design, Lee et al. [58], Oda et al. [59], Stein et al. [60], Rosterman et al. [48], Shetty et al. [61], and Hunt et al. [41] opted for crossover studies, while Jung et al. [62] conducted retrospective analyses. The duration of crossover varied across studies, with Lee et al. [58] and Stein et al. [60] utilizing a 4 h crossover, Oda et al. [59] opting for a 3 h crossover, Hunt et al. [41] conducting a 2 h crossover, and Rosterman et al. [48] implementing a lengthy 12 h crossover. Collectively, the study populations encompass preterm infants under 37 weeks in the study by Lee et al. [58], infants below 30 weeks on invasive ventilation with desaturation events in the study by Oda et al. [59], and low-birth-weight infants in the study by Stein et al. [60]. Additionally, Rosterman et al. [48] studied infants over 22 weeks who were stable on mechanical ventilation, and Hunt et al. [41] examined infants born under 32 weeks and ventilated beyond one week. Shetty et al. [61] analyzed infants under 32 weeks on ventilation for over two weeks, and Jung et al. [62] retrospectively investigated infants under 32 weeks on mechanical ventilation with RSS > 4. Regarding interventions, most studies compared NAVA with Synchronized Intermittent Mandatory Ventilation (SIMV) with Pressure Support (PS) as CMV. Hunt et al. [41] and Shetty et al. [61] employed Assistant Control (AC) or SIMV as interventions for CMV.

### 4.1. Practical Implications and Future Research Directions

According to the systematic review, NAVA may present short-term benefits, including reduced oxygen requirements at 28 days and favorable ventilator parameters such as low PIP and Tidal Volume, as well as decreased Work of Breathing and Respiratory Severity Score. Nevertheless, no statistically significant difference exists in long-term outcomes, particularly regarding BPD. Consequently, additional well-designed large-scale randomized controlled trials are imperative to explore NAVA’s long-term implications.

### 4.2. Limitations

Within this systematic review, it is crucial to acknowledge that certain limitations are inherent. In the evaluation of reporting biases, a funnel plot would have been generated to assess potential publication bias if a sufficient number of trials were eligible for inclusion in the meta-analysis. However, due to the lack of a sufficient number of trials identified for inclusion in the meta-analysis, the creation of a funnel plot to assess publication bias was not pursued. A limited number of studies and the observed heterogeneity among the included studies highlight the need for cautious interpretation. In addition to significant heterogeneity in the baseline patient characteristics among the RCTs included in our review, particularly concerning gestational age and birth weight, these studies employed varied terminology for BPD. Moreover, BPD was treated as a secondary outcome or necessitated multiple linear regression analysis to account for several independent variables. Consequently, when analyzed individually, these outcomes might lack sufficient statistical power, enabling only the generation of hypotheses that should be confirmed in future trials, with BPD being the primary outcome. In light of this, our review aggregated data from these individual trials to enhance the statistical power and the precision of effect size estimates. Moreover, it is essential to mention that only two RCTs were eligible for the review to analyze the primary objective, further highlighting the constraints of available evidence. Due to the limited availability of data, conducting a statistical analysis on other outcomes, such as sedation requirements and neurological outcomes between NAVA and CMV, was not feasible.

## 5. Conclusions

Despite the observed short-term advantages, including lower PIP and MAP, as well as reduced oxygen requirements at 28 days, our study did not find a significant reduction in BPD incidence at 36 weeks PMA with invasive NAVA. Therefore, determining the clinical benefit of NAVA on late neonatal outcomes remains inconclusive based on the current literature. In summary, our analysis, limited by the number of studies, underscores the need for future trials to thoroughly investigate the effects of NAVA on long-term outcomes in premature infants. As a suggestion for future research, it would be beneficial to conduct large-scale studies with a focus on the impact on BPD. Large sample sizes in future research endeavors would provide more power to determine the impact of NAVA on BPD outcomes conclusively.

## Figures and Tables

**Figure 1 healthcare-12-00632-f001:**
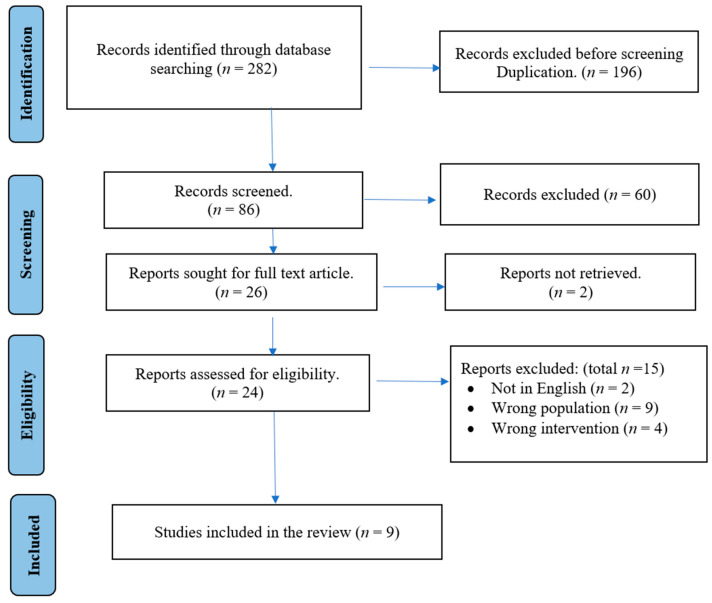
Search Methods.

**Figure 2 healthcare-12-00632-f002:**
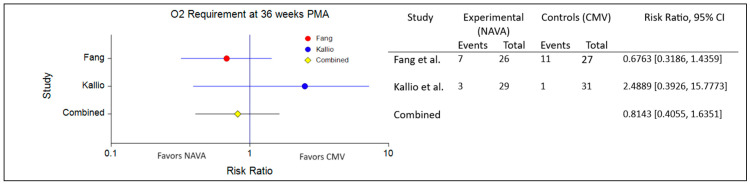
Forest plot for impact on O_2_ requirement at 36 weeks PMA [56,57].

**Figure 3 healthcare-12-00632-f003:**
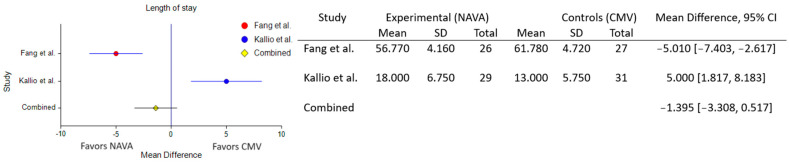
Forest plot for NICU length of stay [56,57].

**Figure 4 healthcare-12-00632-f004:**
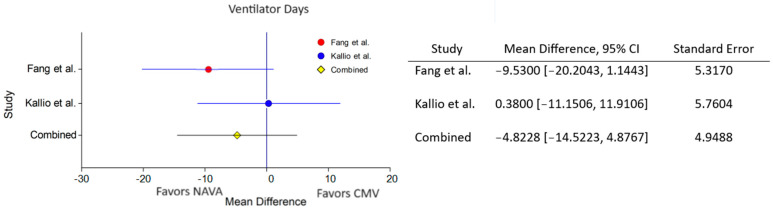
Forest plot for ventilator days [56,57].

**Table 1 healthcare-12-00632-t001:** Characteristics of studies included in the review.

Study, Year	Study Type	Inclusion, Exclusion, and Comparison Criteria	Sample Size	BW (g) Mean ± SD or Median (Range)	GA (Weeks) Median	Enrollment Age Median (Range)
Fang, 2022 [56]	RCT	GA < 32 weeks, intubated for delivery room resuscitation. Exclusion: lethal anomalies, BW < 500 g. Comparison: SIMV or SIMV PS.	53	1207.9 ± 47.2	29.0 ± 0.3	<24 h
Kallio, 2016 [57]	RCT	GA 28–36 6/7 weeks, on invasive ventilation for RDS for at least 4 h. Exclusion: diaphragm defects, inability to insert gastric tube, severe asphyxia, chromosomal abnormalities. Comparison: patient-triggered PC ventilation.	60	1735.9 ± 812	31.6 ± 2.6	9.3 (2.3–49) days
Lee, 2012 [58]	Randomized Crossover	GA < 37 weeks, on invasive ventilation with spontaneous breathing. Exclusion: major anomalies, IVH (grade III+), phrenic nerve palsy. Comparison: SIMV with PS (4 h crossover).	19	1210 (670–2580)	29.1 (25–36.4)	7 (2–70) days
Oda, 2021 [59]	Observational Crossover	GA < 30 weeks, on invasive ventilation with desaturation events. Exclusion: major anomalies. Comparison: SIMV + PS (3 h crossover).	20	610 (400–1160)	26 4/7 (23–29 3/7)	20 (1–82) days
Stein, 2013 [60]	Prospective Crossover	Low-birth-weight infants on invasive ventilation. Comparison: PCV (4 h crossover).	5	697 (370–1140)	26.2 (25–29)	24 (6–34) days
Rosterman, 2018 [48]	Randomized Crossover	GA > 22 weeks, stable on MV. Exclusion: phrenic nerve palsy, respiratory suppression due to sedation or neurologic compromise. Comparison: SIMV (PC)+PS (12 h crossover).	22	734 (432 to 3165)	26 4/7 (23 to 39)	40 (3 to 135) days
Hunt, 2020 [41]	Crossover	Born < 32 weeks, ventilated beyond 1 week. Comparison: A/C or SIMV (2 h crossover).	18	750 (454–950)	25.3 (23.6–30.3)	20.5 (8–58) days
Shetty, 2017 [61]	Crossover	GA < 32 weeks, on invasive ventilation for > 2 weeks. Comparison: A/C or SIMV.	9	750 (545–830)	25 (22–27)	20 (8–84) days
Jung, 2020 [62]	Retrospective	GA < 32 weeks on mechanical ventilation with RSS > 4. Comparison: SIMV-PC (PS)—(pre- and post-NAVA conversion).	29	680 (370–1230)	25.4 (23.4–30.3)	32.1 (26.4–43.3) days

**Table 2 healthcare-12-00632-t002:** Summary of levels of evidence and grades of recommendation.

Study	Level of Evidence	Grade of Recommendation
Fang et al. [56]	1++	A
Kallio et al. [57]	1++	A
Lee et al. [58]	2++	B
Oda et al. [59]	2+	C
Stein et al. [60]	2+	C
Rosterman et al. [48]	2++	B
Hunt et al. [41]	2+	C
Shetty et al. [61]	2+	C
Jung et al. [62]	2+	C

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
