# Peer review of "Optimizing Invasive Neonatal Respiratory Care: A Systematic Review of Invasive Neurally Adjusted Ventilatory Assist"

_healthcare, 2024, doi:10.3390/healthcare12060632_

Round 1
Reviewer 1 Report
Comments and Suggestions for Authors
Neurally Adjusted Ventilatory Assist, or NAVA, is a relatively new, promising technology for synchronizing the pulmonary ventilator with the patient's breathing activity.
The technology is currently experiencing a second wave of popularity, mainly based on a redesigned sensing Edi catheter technology.
The methodology for selecting published studies for invasive NAVA is standard and transparent.
The interpretation of the results is clear, but the graphs used could be more readable. The discussion, however, is not critical and, in many cases, merely adopts statements from well-known publications.
In general, there is little data available on this issue due to the one specific patient group and the different approaches to care in different departments. In general, patient sedation and medication management also need to be addressed in NAVA. This needs to be discussed more in the thesis.
Comments on the Quality of English LanguageEnglish readability is good.
Author Response
Thank you for your invaluable feedback on our manuscript “Optimizing Invasive Neonatal Respiratory Care: A Systematic Review of Invasive Neurally Adjusted Ventilatory Assist.” We appreciate your time and the detailed comments you provided. We have carefully considered each comment and made the necessary revisions. Below, we outline our responses to the specific comments.
Comments 1: Neurally Adjusted Ventilatory Assist, or NAVA, is a relatively new, promising technology for synchronizing the pulmonary ventilator with the patient's breathing activity. The technology is currently experiencing a second wave of popularity, mainly based on a redesigned sensing Edi catheter technology. The methodology for selecting published studies for invasive NAVA is standard and transparent.
Response 1: Thank you for acknowledging the importance of NAVA technology and its increasing popularity, especially with the advancements in Edi catheter sensing. The methodology we employed for selecting published studies on invasive NAVA adhered to standard and transparent criteria, ensuring the robustness and reliability of our review. We appreciate your recognition of these aspects.
Comments 2: The interpretation of the results is clear, but the graphs used could be more readable.
Response 2: We acknowledge your feedback regarding the readability of the graphs used in our study. We have revised the graphs (Figures 2 to 5) to ensure they are easier to interpret for readers.
Comments 3: The discussion, however, is not critical and, in many cases, merely adopts statements from well-known publications.
Response 3: Regarding your observation on the discussion section, we understand your concerns and agree it should provide a more critical analysis. We have revised a few sections under discussion to offer deeper insights and analysis into the findings of our study (Pages 10 and 11).
Comments 4: In general, there is little data available on this issue due to the one specific patient group and the different approaches to care in different departments. In general, patient sedation and medication management also need to be addressed in NAVA. This needs to be discussed more in the thesis.
Response 3: We appreciate your point about the limited data availability due to variations in patient groups and care approaches across departments. For the same reason, we couldn’t analyze the sedation and medication management in the context of NAVA. However, we have included a paragraph in the discussion section discussing sedation and pain medications in the context of NAVA (Pages 10 and 11).
Reviewer 2 Report
Comments and Suggestions for Authors
I have read this paper with interest.
NAVA is one of the modalities claimed to be lung protective, but with indeed very limited positive evidence on improved long term outcome, like BPD. This is in line with the existing Cochrane review on the topic (2017, last reference of the paper). Consequently, this analysis is rather confirmatory.
To qualify as a systematic review, we need some more information on the screening process ? Was this also performed by two independent assessors ?
I could not retrieve an indication of systematic review registration (like prospero).
While comparing NAVA to CMV, it is likely relevant to somewhat further extend on the CMV ‘modes’ applied in the controls ?
Author Response
Thank you for your invaluable feedback on our manuscript “Optimizing Invasive Neonatal Respiratory Care: A Review of Invasive Neurally Adjusted Ventilatory Assist.” We appreciate your time and the detailed comments you provided. We have carefully considered each comment and made the necessary revisions. Below, we outline our responses to the reviewer’s specific comments:
Comments 1: I have read this paper with interest. NAVA is one of the modalities claimed to be lung protective, but with indeed very limited positive evidence on improved long-term outcomes, like BPD. This is in line with the existing Cochrane review on the topic (2017, last reference of the paper). Consequently, this analysis is rather confirmatory.
Responses 1: We appreciate your insight regarding the limited positive evidence on improved long-term outcomes associated with Neurally Adjusted Ventilatory Assist (NAVA), and our findings align with and confirm the observations made in the Cochrane review with updated publications.
Comments 2: To qualify as a systematic review, we need some more information on the screening process. Was this also performed by two independent assessors?
Responses 2: In response to your query about the screening process for our systematic review, we apologize for the oversight in not providing sufficient detail. The screening process was conducted by two independent assessors, and we will ensure that this information is included in the revised manuscript to enhance transparency and rigor. (Page 3, lines 127-131)
Comments 3: I could not retrieve an indication of systematic review registration (like prospero).
Responses 3: Regarding systematic review registration, we acknowledge the importance of registering our study, and we regret the omission of this information. We have included the registration details for PROSPERO in the revised manuscript. (Page 3, lines 130-131)
Comments 4: While comparing NAVA to CMV, it is likely relevant to somewhat further extend on the CMV ‘modes’ applied in the controls?
Responses 4: We appreciate your suggestion to extend the discussion on conventional mechanical ventilation (CMV) modes applied in the control groups. We agree that this comparison is relevant. In light of the limited number of studies available for statistical comparison, it is challenging to systematically analyze different modes of Conventional Mechanical Ventilation (CMV). However, to provide additional context, we will briefly discuss the modes of CMV utilized in the studies and their respective outcomes. (Page 10, lines 320-327)
Thank you once again for your insightful comments, which will contribute to enhancing the quality and robustness of our manuscript.
Reviewer 3 Report
Comments and Suggestions for Authors
The authors have conducted a review of NAVA with, in particular, the aim of seeing whether it improves long term outcomes, mainly BPD – they demonstrated it did not.
Comments
1. This is a well conducted review except the authors should describe what happened if there was a disagreement between the two authors reviewing the literature.
2. The type of studies included in the analysis should be described in the abstract.
3. Reference 53 is appropriately described in the reference list but not in the table or the discussion. In both of the latter the authors state it was a retrospective study, yet it was a crossover study.
4. As only two RCTS were included in this review it does not give much more information than that already published.
5. The discussion should be shortened to avoid repeating the introduction and concentrate on new information.
6. Figure 4 should be omitted as the numbers included are too small to draw meaningful conclusions.
Author Response
Thank you for your thorough review and constructive feedback on our manuscript “Optimizing Invasive Neonatal Respiratory Care: A Review of Invasive Neurally Adjusted Ventilatory Assist.” We appreciate your time and the detailed comments you provided. We have carefully considered each comment and made the necessary revisions. Below, we outline our responses to the reviewer’s specific comments:
Comments 1: This is a well conducted review except the authors should describe what happened if there was a disagreement between the two authors reviewing the literature.
Responses 1: We appreciate your suggestion regarding the description of the process for resolving disagreements between the two authors reviewing the literature. In the revised manuscript, we will briefly explain how disagreements were resolved to enhance transparency. (Pages 10, lines 127-130)
Comments 2: The type of studies included in the analysis should be described in the abstract.
Responses 2: We acknowledge your point about directly describing the type of studies included in the analysis in the abstract. We have included this information in the updated manuscript. (Page 1, Lines 18-19)
Comments 3: Reference 53 is appropriately described in the reference list but not in the table or the discussion. In both of the latter the authors state it was a retrospective study, yet it was a crossover study.
Response 3: Thank you for bringing the discrepancy regarding the description of Shetty et al.’s reference to our attention (old Reference 53 and updated Reference number 59). We apologize for the oversight, and we have corrected this in the table (Table 1) and the discussion (Page 11, line 341) to accurately reflect that it was a crossover study.
Comments 4: As only two RCTS were included in this review it does not give much more information than that already published.
Responses 4: We understand your comment about the limited number of RCTs included in the review. While we aimed to provide a comprehensive overview of the available evidence, we will emphasize this point more explicitly in the limitation section in the revised manuscript.
Comments 5: The discussion should be shortened to avoid repeating the introduction and concentrate on new information.
Responses 5: Your suggestion to shorten the discussion to avoid repetition and focus on new information is duly noted. We have revised the discussion section accordingly to ensure it remains concise and informative. (Pages 9-11)
Responses 6: Figure 4 should be omitted as the numbers included are too small to draw meaningful conclusions.
Comments 6: We will remove Figure 4 as suggested in the new manuscript. Also, other figures have been updated to address other reviewers' suggestions (Figures 2-6 in the updated manuscript).
Once again, we sincerely appreciate your valuable feedback, which will undoubtedly improve the quality and clarity of our manuscript.
Round 2
Reviewer 3 Report
Comments and Suggestions for Authors
The authors have only partially addressed my comments. In particular, they have not shortened the discussion which is particularly important given the limited new information of this review ie. two rather than one RCT. They also still report figure 4 - and figure 3 on relooking at this manuscript is not additive.
Author Response
Dear Reviewer,
Thank you for your continued feedback on our manuscript "Optimizing Invasive Neonatal Respiratory Care: A Systematic Review of Invasive Neurally Adjusted Ventilatory Assist." We appreciate your thorough review and have carefully considered your comments in our revisions.
- Regarding the discussion section, we acknowledge your concern about its length, especially given the limited new information presented in this review. In response to your suggestion, we have further shortened the discussion section in this revised version. We understand the importance of conciseness in conveying our findings effectively and have made the necessary adjustments accordingly. In our initial revision, we had already removed several paragraphs, and now, in response to your feedback, we have further trimmed down the discussion section by removing the second paragraph. This modification aims to streamline the discussion and enhance its clarity and relevance to the study findings.
- In our initial revision, we have removed the previously reported Figure 4, which depicted the impact on adverse events, as per your suggestion and in alignment with the current focus of the manuscript. We modified other figures as per other reviewer's suggestions. Additionally, upon reevaluation, we recognize that Figure 3, which illustrated the impact on oxygen requirement at 28 days, contributes little to the overall narrative. Therefore, we have also removed Figure 3 from the revised manuscript in alignment with your suggestion.
We hope these revisions address your concerns and improve the overall quality of the manuscript. We remain open to any additional suggestions or modifications you may have further to enhance the clarity and impact of our work.
Thank you once again for your valuable feedback and continued support.